# Stability of the Inclusion Complexes of Dodecanoic Acid with α-Cyclodextrin, β-Cyclodextrin and 2-HP-β-Cyclodextrin

**DOI:** 10.3390/molecules28073113

**Published:** 2023-03-30

**Authors:** Zdzisław Kinart

**Affiliations:** Department of Physical Chemistry, Faculty of Chemistry, University of Lodz, Pomorska 163/165, 90-236 Lodz, Poland; zdzislaw.kinart@chemia.uni.lodz.pl

**Keywords:** electric conductivities, α-cyclodextrin, β-cyclodextrin, 2-HP-β-cyclodextrin, aqueous solutions of sodium salts of dodecanoic acid, complex constants, thermodynamic function

## Abstract

In the presented work, the stability of the formation of inclusion complexes of dodecanoic acid (lauric acid) with three cyclodextrins, α-cyclodextrin, β-cyclodextrin and 2-HP-β-cyclodextrin, was analyzed from the point of view of the size of the cavity in cyclodextrins, their molar mass and the structure of the studied fatty acid. The measurements were made in a wide temperature range of 283.15–318.15K. The conductometric method was used for these studies. The results obtained allowed us to determine the value of the theoretical limiting molar conductivity ($\Lambda_{m}^{0}$) of the studied complexes, the values of the inclusion complex formation constants (*K_f_*) and the values of thermodynamic functions (Δ*G*^0^, Δ*H*^0^ and Δ*S*^0^) describing the complexation process in the studied temperature range.

## 1. Introduction

The purpose of the presented work is to analyze and compare the complexation constants of dodecanoic acid (lauric acid) by α-cyclodextrin, β-cyclodextrin and 2-HP-β-cyclodextrin in a wide temperature range of 283.15–318.15 K. It has been shown that by using conductometric measurements, it is possible to trace the process of formation of inclusion complexes in a very precise way, ad to determine the values of the formation constants of these complexes, *K_f_*, and to evaluate the values of changes in thermodynamic functions that describe this process as a function of temperature in an aqueous solvent.

The structure of cyclodextrin and its physicochemical and biological properties contributed to the wide use of these compounds in drug research. Cyclodextrins are natural cyclic oligosaccharides composed of α-D-glucopyranose molecules linked by an α-1,4-glycosidic bond and are obtained as a result of starch enzyme degradation [1]. These compounds, whose molecules are made up of six, seven, and eight residues of α-D glucopyranose, are called α-cyclodextrin, β-cyclodextrin and γ-cyclodextrin, respectively [2,3,4]. The structure of cyclodextrin is shown in Figure 1:

The most common cyclodextrins are α-, β- and γ cyclodextrin. These are crystalline compounds and, as mentioned above, are constructed of 6, 7 and 8 glucose residues, respectively, linked by an α-1,4-glycosidic bond.

The hydrophobic/hydrophilic nature of cyclodextrin molecules gives them the ability to recognize molecular complexes as a result of the formation of host–guest complexes with a whole range of reagents. In these complexes, the “guest” molecule is placed in the cavity of the “host” molecule, cyclodextrin [3]. The formation of inclusion complexes depends on various factors.

First is the charge and nature of the ligand and the nature of noncovalent mutual interactions of substrate–ligand type [7]. Only a “guest” molecule having the right shape and size can be included in the cavity of the cyclodextrin. We deal with the phenomenon of partial inclusion when the “guest” molecule cannot fully penetrate the interior of the “host”. Then, the hydrophobic groups of the molecule may be present in the cyclodextrin cavity, and the hydrophilic groups remain outside and can combine with solvent molecules and hydrophilic cyclodextrin groups [6,8]. The structure of cyclodextrins and their physicochemical and biological properties make them very useful in many areas of human life. These compounds are widely used in medicine, pharmacy, chemistry, agriculture, etc. The study of the mechanism of inclusion of various molecules within cyclodextrins plays an important role in research in the field of supramolecular chemistry [9]. Using modern microscopic techniques (e.g., scanning electron microscopy, SEM), it is possible to study the morphology and structure of cyclodextrin inclusion complexes obtained in the solid phase.

Cyclodextrins willfully form inclusion complexes with biologically active substances (selected drugs) as stabilizing systems. New-generation antidepressants and neuroleptics, as well as fluoroquinolone antibiotics in the form of complexes with cyclodextrins, are known. The active compounds of these drugs are sensitive to the action of various oxidizing, hydrolyzing and polymerizing agents, which adversely affect the physicochemical properties of the active substances and their therapeutic effect. Cyclodextrins are also used in biomedicine for the production of hydrogels used as an effective and safe drug delivery system [10,11,12,13].

Dodecanoic acid (lauric acid C_12_H_24_O_2_) is a representative of saturated fatty acids. It is naturally present in mothers’ milk and coconut oil (approximately 50% of its content), as well as in laurel fruit. It got its name from bay leaves. Numerous studies have shown that lauric acid has antibacterial, antiviral and antifungal properties. Monolaurin is a safe and non-toxic substance and is therefore widely used in many industries. The use of lauric acid is very wide. The benefits of this compound are used, among others, by the cosmetic and dermatological industries (for the production of deodorants, cosmetics, insecticides and detergents). It is also used in the food and beverage industry for food production as an emulsifier and preservative. Additionally, it prevents infections of the digestive system. It is also of great importance in the pharmaceutical industry (with weakened immunity as a natural antibiotic that supports the immune system). Nature has placed this compound in mothers’ milk. Studies on this compound show that it has a positive effect on the level of good HDL cholesterol. When used as a supplement, it also supports the circulatory system. Lauric acid also has antiviral properties. Monolaurin is effective against lipid membrane viruses (such as HIV-1, measles, sarcoma and vesicular stomatitis) [14,15,16].

The studies carried out in this work enable a deeper understanding of the mechanisms of inclusion interactions of one fatty acid with cyclodextrins and the analysis of the impact of the spatial structure and the size of the “cavity” of the tested cyclodextrins on the stability of the inclusion complexes formed by them with dodecanoic acid.

## 2. Results

Basic values of the density, viscosity and relative permittivity of water (which was used as a solvent in this work) are summarized in Appendix A. These values were used for calculations in the subsequent stages of determining the parameters tested. The values of molar conductivities and molar concentrations for the sodium salt of the tested fatty acid with α-cyclodextrin, β-cyclodextrin and for 2-HP-β−cyclodextrin in the temperature ranges are presented in Appendix A.

In this paper, α-cyclodextrin, β-cyclodextrin and 2-HP-β-cyclodextrin (CD) form inclusion complexes with dodecanoic acid (Dod^−^) with a 1:1 stoichiometry:$$\mathrm{CD}+{\mathrm{Dod}}^{-}⟷{\mathrm{DodCD}}^{-}$$

The formation constant (*K_f_*) of these complexes has the form:(1)$$K_{f}=\frac{\left[{DodCD}^{-}\right]}{\left[{Dod}^{-}][CD\right]}$$

The results obtained from the conductometric measurements allow one to determine the values of these constants.

The molar conductivity of the sodium salt of a selected phenolic acid can be expressed by the molar conductivities of the acid salt with an uncomplexed anion (*Λ_NaDod_*) and the alkaline salt of the acid studied, where the anion forms an inclusion complex with cyclodextrin (*Λ_CDNaDod_*).

The molar conductivity of a solution is represented by the following relationship:(2)$$\Lambda =\frac{1000\cdot \kappa }{C_{NaDod}}$$
where:

*κ*—specific conductivity in the unit [S·m^−1^];

*C_NaDod_*—molar concentration of sodium salts of phenolic acid.

The molar conductivity of the tested dodecanoic acid, calculated from Equation (2), can be expressed by the molar conductivities of the tested sodium salt of the studied acid with an uncomplexed anion (*Λ_NaDod_*) and sodium salt of this acid in which the anion forms an inclusion complex with cyclodextrin (*Λ_CDNaDod_*).

The molar conductivity and the formation constant (*K_f_*) values of inclusion complexes were calculated according to the procedures described in our previous work [17,18]. Taking this into account and the calculation procedures described in work [17,18], we obtain the following relations:(3)$$K_{f}\cdot [CD{]}^{2}+\left[K_{f}\cdot \left(C_{DodNa}-C_{CD}\right)+1\right]\cdot \left[CD\right]-C_{CD}=0$$

Taking this into account, the equation for the molar conductivity of the solution before adding the cyclodextrin takes the form:(4)$$\begin{array}{l} \Lambda =\left[K_{f}\cdot \left(c_{DodNa}-C_{CD}\right)-1+\sqrt{K_{f}^{2}\cdot (C_{CD}-C_{DodNa}{)}^{2}+2K_{f}\cdot \left(C_{DodNa}+C_{CD}\right)+1}\right]\\ \cdot \left[\frac{\Lambda_{NaDod}-\Lambda_{CDNaDod}}{2K_{f}{\cdot C}_{DodNa}}\right]+\Lambda_{NaDodCD} \end{array}$$

A review of the literature data [17,18] shows that the values of *Λ_NaDod_* and *Λ_NaDodCD_* may change, and their values can be described by the following equations:(5)$$\Lambda_{NaDod}=\Lambda_{oNaDod}-S\cdot c_{NaDod}^{1/2}+E\cdot c_{NaDod}\cdot lnc_{NaDod}+J_{1}\cdot c_{S}+J_{2}\cdot c_{NaDod}^{3/2}$$
(6)$$\Lambda_{NaDodCD}=\Lambda_{oCDNaDod}-S\cdot c_{NaDod}^{1/2}+E\cdot c_{NaDod}\cdot lnc_{NaDod}+J_{1}\cdot c_{S}+J_{2}\;\cdot c_{NaDod}^{3/2}$$

The values of the parameters *S*, *E*, *J*_1_, and *J*_2_ are related to relaxation and electrophoretic effects. However, it should be remembered that the values of *J*_1_ and *J*_2_ depend on the so-called parameter of maximum ion approximation.

## 3. Discussion

As you can see, the presented equations allow for the assessment of the fixed values of the creation of inclusive complexes and theoretical conductivity (which was presented in this work). The research presented in this work also allowed for the analysis of the correctness of the use of these equations from the point of view of the size of cyclodextrin and ions that make up complexes with them.

The functions discussed have also been repeatedly analyzed by other authors and were described in the literature [19,20].

Appendix A summarize the values of the total molar concentration of sodium dodecanoate and the concentration of the ligand, as well as the values of determined molar conductivities for the complexes of α-cyclodextrin, β-cyclodextrin and 2-hydroxypropyl-β-cyclodextrin with a dodecanoate ion in the studied temperature range of 283.15–318.15 K. When the above values were met, it was found that the increase in temperature caused an increase in the molar conductivity of the salt, which was associated with a decrease in viscosity, and, as a result, a higher speed of movement of ions in the tested solvent.

The dependences of the molar conductivity on the concentration of the ligands (α-cyclodextrin, β-cyclodextrin and 2-hydroxypropyl-β-cyclodextrin) in the tested temperature range are presented in Appendix A. As can be seen, the molar conductivity of the solution decreased with increasing ligand concentration, indicating that as more cyclodextrin was added, more and more dodecanoate ions were included, and thus, the conductivity of the solution decreased. Once the ligand was present in the amount corresponding to the salt content of the tested acid, the molar conductance values changed considerably less. As can be seen, an increase in the molar conductances was also visible with an increase in the molar mass (size) of the tested cyclodextrins. These values, as mentioned above, decreased with increasing molar concentration but also increased monotonically as a function of temperature for all studied cyclodextrins, which is consistent with the theory of molar conductivity.

Figure 2 shows the changes in the inclusion complex (*K_f_*) as a function of T [K] for α-cyclodextrin, β-cyclodextrin and 2-hydroxypropyl-β-cyclodextrin in the investigated temperature range. These figures and the data contained in Table 1 show that in the case of α- and β-cyclodextrin, the constant values of the formation of the inclusion complex decrease with increasing temperature.

In the case of 2-HP-β-cyclodextrin, changes in inclusion complex formation constants as a function of temperature changes are more complex. Initially, as the temperature increased, the *K_f_* values increased, and after reaching a maximum at about 298.15K, they began to decrease. It is also worth mentioning that, in general, significantly higher values of the formation constants were observed for α and β-cyclodextrin than for 2-hydroxypropyl-β-cyclodextrin. The value of the inclusion complex formation constant was influenced by the size of the cyclodextrin gap and the length of the carboxylic chain. Higher values of the constant may have indicated better matching of the particle geometry to the size of the gap. The presence of hydroxyisopropyl groups in cyclodextrin also appeared to be significant. This presence dramatically improved the solubility of cyclodextrin in water but may have hindered the formation of an inclusion complex with hydrophobic hydrocarbon chains. When comparing the *K_f_* values presented in Table 1, it can be observed that the *K_f_* values of the acid inclusion complexes studied with both α- and β-cyclodextrin were much higher than those of the acid inclusion complexes of this acid with 2-HP-β-cyclodextrin. Most likely, the increase in the length of the hydrocarbon chain increased the number of possible conformers in the cyclodextrin cavity, which was reflected in the larger formation constants. Our conclusions are confirmed by data from the compound literature related to complexation with the general formula CH_3_(CH_2_)_n_X by cyclodextrins, where X = CH_3_, COOH, COO^−^, OH^−^ [21,22,23,24]. However, there is no doubt that, taking into account the data of our research and the literature, the values of inclusion complex formation constants of various cyclodextrins with the same “guest” are higher for α- and β-cyclodextrin than for complexes with 2-HP-β-cyclodextrin. This indicates a very high dependence of the *K_f_* value on the size of the cavity of cyclodextrin. Taking into account the size of the cavity diameter of the tested cyclodextrins, which was approximately 5.3 Å for α-cyclodextrin, approximately 6.3 Å for β- cyclodextrin and approximately 11.6 Å for 2-HP-β-cyclodextrin, it could be assumed that the dodecanoate molecule had much more free space in the cavity 2-HP-β-cyclodextrin and may therefore have formed less complexes. An additional factor influencing the differences discussed in *K_f_* values may be the fact that the dodecanoate molecules placed in the 2-HP-β-cyclodextrin cavity could interact with its hydroxyisopropyl groups. The existing differences between the *K_f_* values presented in this work and the values presented in the literature for α-cyclodextrin and β-cyclodextrin (see Table 1) probably result from differences in the solvent compositions and the analysis of different temperatures. Georgiju showed in his work [25] that different basic concentrations of the dodecanoic acid anion can cause fluctuations in the *K_f_* value. The dependence of the theoretical Λ as a function of changes in temperature (T [K]) for the systems tested is shown in Figure 3.

As can be seen, these values increased with an increase in temperature (which is an obvious effect). It is worth emphasizing, however, that these values were definitely lower than the conductivity values of the tested salt when the anion was not complexed. This is obviously related to the much larger size of the complexed anion.

The review of data from the literature indicates a lack of any information regarding the values of the constants that create inclusion complexes of dodecanoic acid with the studied cyclodextrins. In the literature, the values of *K_f_* are presented but only in relation to lower homologists of fatty acids with cyclodextrins. These results indicate large discrepancies in the cited *K_f_* values depending on the measuring methods used by various authors (see Table 2):

The temperature dependences of the complex formation constant were used to determine the free enthalpy of the complex formation.
(7)$$∆G\left(T\right)=-RTlnK_{f}(T)$$

The above relationship can be presented as follows:(8)$$∆G\left(T\right)=A+BT+CT^{2}$$

The values of parameters *A*, *B* and *C* are calculated from the figures showing the changes in the Gibbs free enthalpy as a function of temperature and then are inserted into the analyzed equation [27,28,29]. The values of these parameters for the aspect studied (in this paper, systems) are summarized in Appendix A.

The enthalpy values are presented as the first derivatives of the free enthalpy after temperature at constant pressure.
(9)$$∆S^{0}=-(\frac{\partial ∆G^{0}}{\partial T}{)}_{p}=-B-2CT$$

The enthalpy is calculated from the following relationship:(10)$$∆H^{0}=∆G^{0}+T∆S^{0}=A-CT^{2}$$

The determined values of the analyzed thermodynamic functions are listed in Table 3.

As can be seen, the ∆*G*^0^ values for 2-hydroxypropyl-β-cyclodextrin were definitely less negative than those for α- and β-cyclodextrin. This proves greater spontaneity in the process of formation of inclusion complexes of dodecanoate anions with α- and β-cyclodextrin. The courses in changes in the thermodynamic functions are presented in Figure 4.

In the case of entropy and enthalpy of formation, the nature of their changes, as well as their values as a function of temperature changes, differed slightly for the discussed cyclodextrins, exactly as can be seen in Figure 5 and Figure 6.

In the case of α- and β-cyclodextrin, the enthalpies of formation were negative over the entire temperature range. However, in the case of 2-HP-β-cyclodextrin, the formation process became exothermic only for temperatures greater than 298.15K. The entropy of complex formation also changed with increasing temperature for all of the discussed cyclodextrins. However, for 2-HP-β-cyclodextrin, positive values prevailed at lower temperatures. Only at temperatures higher than 310.15K did they become negative. For complexes of α and β-cyclodextrin, the entropy values became negative with a temperature of 300.15K. The observed differences may have resulted from not only the different structures of the cyclodextrins discussed, but also from different amounts of water molecules that were located in the “cavities” of the studied cyclodextrins. Thus, the complexation process is a quite complex phenomenon consisting of the effects of dehydration of the dodecanoic anion, its placement inside the cyclodextrin and removal of water from the inside of the cyclodextrin.

## 4. Materials and Methods

### 4.1. Materials

High-purity dodecanoic (lauric acid) acid, α-cyclodextrin, β-cyclodextrin and 2-HP-β-cyclodextrin were used. All information regarding their purity and suppliers is presented in Table 4.

### 4.2. Characterization Methods

All tested solutions were prepared using the gravimetric method in order to avoid measurement errors. Dodecanoic acid sodium salt was obtained by mixing dodecanoic acid and aqueous sodium hydroxide in a stoichiometric ratio of 1:1. The mixture was then heated and stirred until the acid dissolved and the solvent evaporated. The salt obtained in this way was recrystallized twice from aqueous ethanol solutions and then dried using a vacuum dryer. A detailed description for the salt method of obtaining the tested and other sodium salts of fatty acids is presented in the works: [30,31,32]. Conductometric measurements were made using a conductometric bridge of the highest-accuracy-class RLC Wayne-Kerr 6430B with an uncertainty of 0.02%. The measuring cell was a three-electrode cell made of Pyrex glass, which did not contain sodium, which would interfere with the measurement. The electrodes in this cell were made of platinum. The measuring cell in the tested temperature range of 283.15–318.15 K was calibrated using KCl as a reference substance. Calibration ensured that at a given temperature, there would be no measurement errors resulting from changes in the deviations of the electrodes and the glass at the tested temperatures. All measurements were made at different frequencies, v = 0.2, 0.5, 1.0, 1.5, 2.0, 3.0, 5.0, 10.0 and 20.0 kHz, to obtain the most accurate conductivity value molar. All measured conductivity values, *λ = 1/R_∞_,* were the results of the extrapolation of cell resistance, *R_∞_*(*ν*), to infinite frequency *R_∞_ = lim_ν_^®^_∞_R*(*ν*) using the empirical function *R*(*ν*) = *R_∞_ + A/ν* (where parameter *A* is specific for the cell) [33,34]. Taking into account the sources of errors (calibration, sample purity and measurements), the estimated uncertainty of the measured conductivity values was estimated at ±0.05%.

## 5. Conclusions

A review of data from the literature shows that in this work, the molar conductivity measurement technique and innovative computational methods using modified equations were used for the first time to determine the values of the inclusion complex formation constants (*K_f_*) in dodecanoic acid with α-, β- and 2-HP-β-cyclodextrin. It has been shown that α-cyclodextrin and β-cyclodextrin form stronger and more stable inclusion complexes compared to 2-HP-β-cyclodextrin. It was found that the difference in the cavity size in cyclodextrin plays a greater role in the complexation process than the length of the carboxylic acid chain. The results obtained in this work also indicate a large impact of the hydrophobicity effects on the complexation process in the systems studied. Taking into account and comparing the results presented in this work (as well as the results contained in our previous work [17,18]) with the literature data presenting the constant values for the formation of inclusion complexes of various carboxylic acids with cyclodextrins obtained by various measurement techniques (mainly spectral), it should be stated that conductometry is the most accurate method used in this type of research. This research technique also allows for the study of the complexation process in a wide range of temperatures. There is no doubt that the conductometric method proposed by us for the determination of the inclusion complex formation constants should be an interesting alternative to other techniques for the study of these effects.

## Figures and Tables

**Figure 1 molecules-28-03113-f001:**
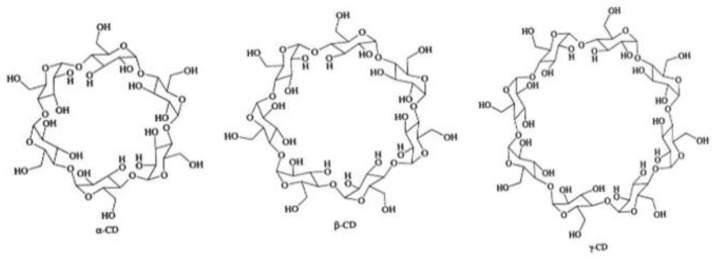
The structure of the cyclodextrin molecule resembles a torus, the inner part of which is hydrophobic, while the outer part is hydrophilic [5,6].

**Figure 2 molecules-28-03113-f002:**
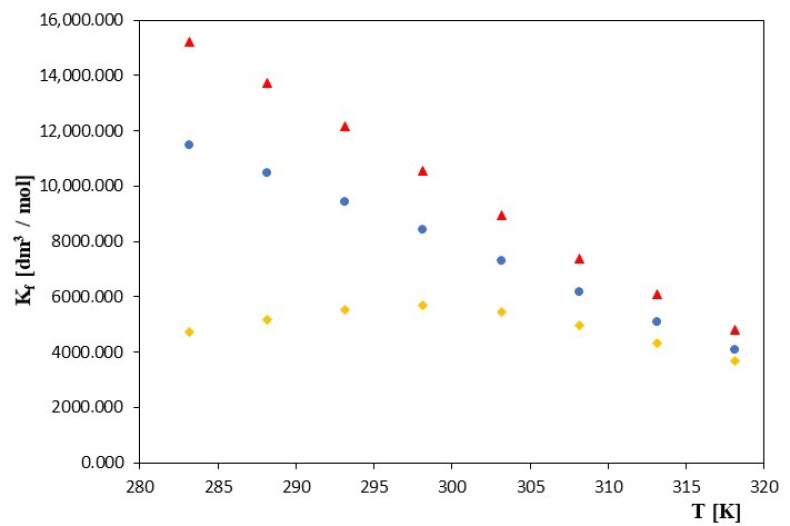
The plots of dependence of the *K_f_* [dm^3^/mol] formation constant in the T [K] function for ▲—α-cyclodextrin, ●—β-cyclodextrin and ♦—2-HP-β-cyclodextrin with all studied salts of dodecanoic acid.

**Figure 3 molecules-28-03113-f003:**
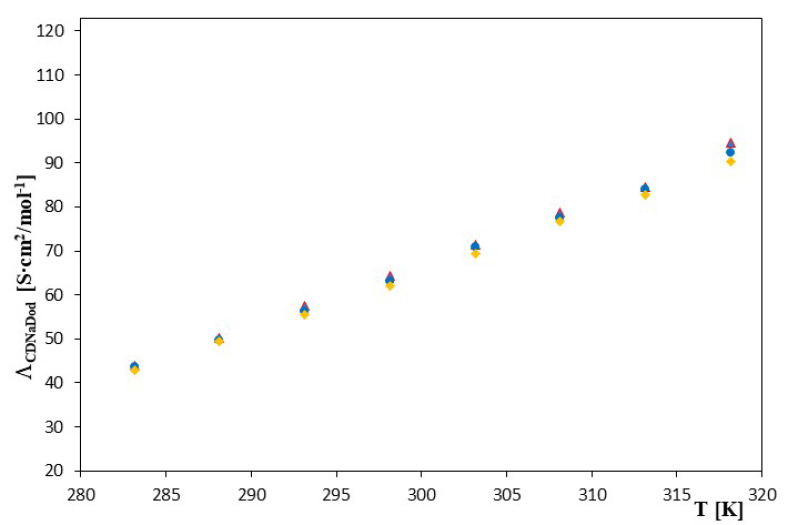
The plots of the dependence of the theoretical conductivity *Λ_CDNaDod_* [S·cm^2^/mol^−1^] in the function T [K] for ▲—α-cyclodextrin, ●—β-cyclodextrin and ♦—2-HP-β-cyclodextrin with all studied salts of dodecanoic acid.

**Figure 4 molecules-28-03113-f004:**
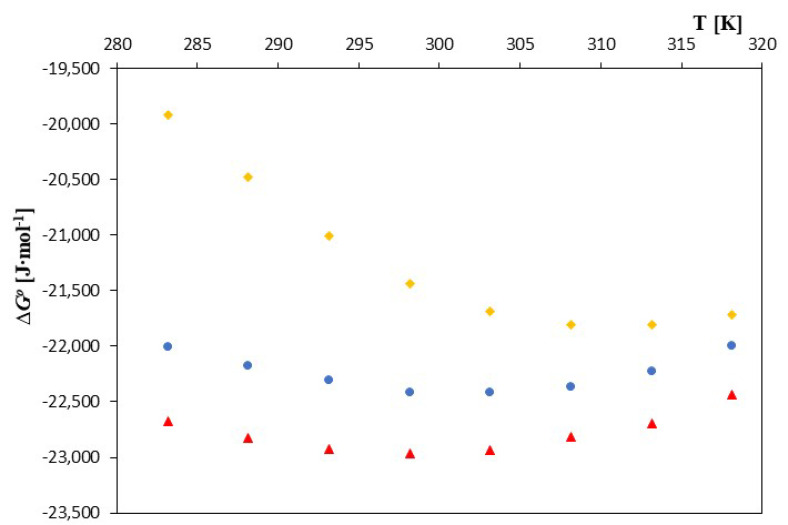
The plots of dependence *∆G^0^* [J·mol^−1^] in the function T [K] for ▲—α-cyclodextrin, ●—β-cyclodextrin and ♦—2-HP-β-cyclodextrin with all studied salt of dodecanoic acid.

**Figure 5 molecules-28-03113-f005:**
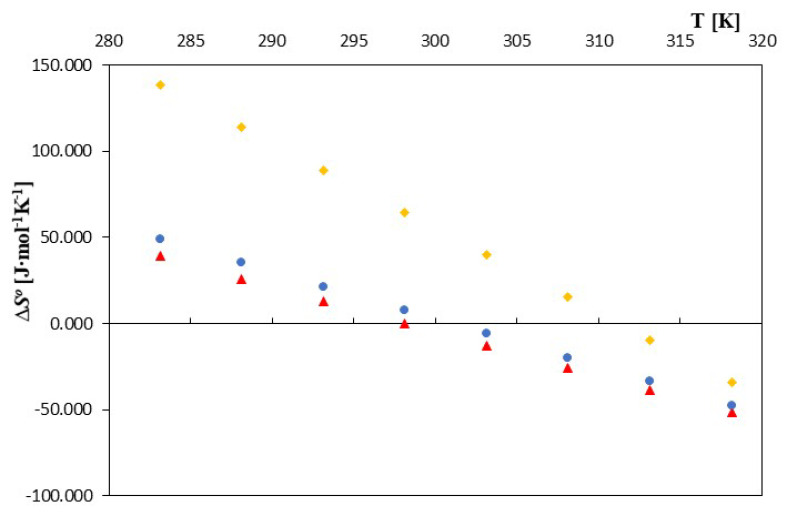
The plots of the dependence *∆S*^0^ [J·mol^−1^‧K^−1^] in the function T [K] ▲—α-cyclodextrin, ●—β-cyclodextrin and ♦—2-HP-β-cyclodextrin with all studied salts of dodecanoic acid.

**Figure 6 molecules-28-03113-f006:**
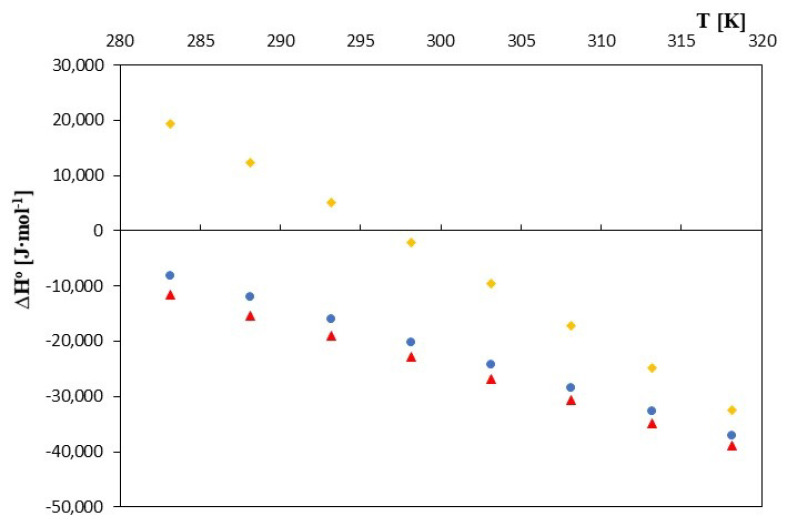
The plots of dependence *∆H*^0^ [J·mol^−1^] in the function T [K] for ▲—α-cyclodextrin, ●—β-cyclodextrin and ♦—2-HP-β-cyclodextrin with all studied salt of dodecanoic acid.

**Table 1 molecules-28-03113-t001:** The value of constant formation *K_f_* [dm^3^/mol] and theoretical conductivity *Λ_CDNaDod_* [S·cm^2^/mol^−1^] for β-cyclodextrin and 2-HP-β-cyclodextrin with the salt of *trans*-caffeic acid.

α-Cyclodextrin					β-Cyclodextrin			
T [K]	*K_f_* [dm^3^/mol]	ln*K_f_* [dm^3^/mol]	*Λ_CDNaDod_* [S·cm^2^/mol^−1^]	σ(Λ)	*K_f_* [dm^3^/mol]	ln*K_f_* [dm^3^/mol]	*Λ_CDNaDod_* [S·cm^2^/mol^−1^]	σ(Λ)
283.15	15,225 ± 8	15,225	44.02 ± 0.01	0.01	11,480 ± 9	9.3484	43.68 ± 0.02	0.01
288.15	13,730 ± 5	13,730	50.22 ± 0.01	0.01	10,466 ± 8	9.2559	49.82 ± 0.01	0.01
293.15	12,150 ± 6	12,150	57.52 ± 0.01	0.02	9449 ± 6	9.1537	56.38 ± 0.01	0.01
298.15	10,555 ± 3	10,555	64.31± 0.01	0.02	8450 ± 4	9.0419	63.00 ± 0.01	0.02
303.15	8940 ± 2	8940	71.41 ± 0.01	0.01	7300 ± 2	8.8956	77.92 ± 0.01	0.01
308.15	7385 ± 1	7385	78.64 ± 0.02	0.01	6180 ± 1	8.7291	77.51 ± 0.01	0.01
313.15	6110 ± 1	6110	84.63 ± 0.01	0.01	5100 ± 1	8.5370	83.96 ± 0.01	0.02
318.15	4825 ± 0.9	4825	94.74 ± 0.02	0.01	4085 ± 1	8.3151	92.25 ± 0.01	0.01
2-HP-β-cyclodextrin								
T [K]	*K_f_* [dm^3^/mol]	ln*K_f_* [dm^3^/mol]			*Λ_CDNaDod_* [S·cm^2^/mol^−1^]	σ(Λ)		
283.15	4719 ± 8	8.4594			43.01 ± 0.01	0.01		
288.15	5161 ± 8	8.5489			49.43 ± 0.01	0.01		
293.15	5531 ± 7	8.6181			55.40 ± 0.01	0.02		
298.15	5697 ± 5	8.6477			62.03 ± 0.02	0.01		
303.15	5455 ± 4	8.6043			69.39 ± 0.01	0.01		
308.15	4981 ± 2	8.5134			76.73 ± 0.01	0.01		
313.15	4335 ± 2	8.3745			82.87 ± 0.01	0.01		
318.15	3680 ± 1	8.2107			90.34 ± 0.01	0.02		

**Table 2 molecules-28-03113-t002:** Literature values of *K_f_* for fatty acids with a carbon chain length of C8 and C10 determined by various physicochemical methods at 298.15K.

System	*K_f_* [dm^3^/mol]
βCD–C10	8000 ± 20 [26]; 3800 ± 100 [25]; 5100 ± 600 [21]
βCD–C8	700 ± 290 [26]; 480 ± 50 [25]; 660 ± 80 [21]

**Table 3 molecules-28-03113-t003:** The values of thermodynamic functions ∆*G*^0^, ∆*S*^0^ and ∆*H*^0^ for α-cyclodextrin, β-cyclodextrin and 2-HP-β-cyclodextrin with the salt of dodecanoic acid.

α-Cyclodextrin				β-Cyclodextrin		
T [K]	∆*G*^0^ [J·mol^−1^]	∆*S*^0^ [J·mol^−1^K^−1^]	∆*H*^0^ [J·mol^−1^]	∆*G*^0^ [J·mol^−1^]	∆*S*^0^ [J·mol^−1^‧K^−1^]	∆*H*^0^ [J·mol^−1^]
283.15	−22671.705	39.072	−11608.847	−22007.063	48.971	−8140.868
288.15	−22824.446	26.116	−15299.201	−22174.135	35.211	−12028.028
293.15	−22922.534	13.160	−19064.762	−22309.760	21.451	−16021.341
298.15	−22964.660	0.204	−22903.921	−22413.288	7.691	−20120.157
303.15	−22931.233	−12.752	−26797.087	−22420.448	−6.069	−24260.205
308.15	−22819.898	−25.708	−30741.904	−22363.529	−19.829	−28473.774
313.15	−22696.739	−38.664	−34804.458	−22226.317	−33.589	−32744.650
318.15	−22434.583	−51.620	−38857.575	−21994.203	−47.349	−37058.223
2-HP-β-cyclodextrin						
T [K]	∆*G*^0^ [J·mol^−1^]		∆*S*^0^ [J·mol^−1^‧K^−1^]		∆*H*^0^ [J·mol^−1^]	
283.15	−19914.238		138.450		19287.897	
288.15	−20480.387		113.812		12314.558	
293.15	−21004.515		89.174		5136.861	
298.15	−21436.072		64.536		−2194.645	
303.15	−21686.153		39.898		−9591.057	
308.15	−21810.947		15.260		−17108.559	
313.15	−21803.194		−9.378		−24739.896	
318.15	−21718.031		−34.016		−32540.202	

**Table 4 molecules-28-03113-t004:** Specification of chemical samples.

Chemical Name	Chemical Formula	Chemical Formula	Source	CAS No	Mass Fraction Purity
Dodecanoic acid (Lauric acid)	C_11_H_21_O_2_	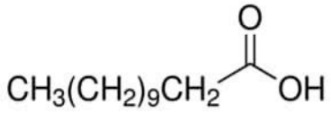	Merck	143-07-7	≥0.999
α -cyclodextrin	C_36_H_60_O_30_	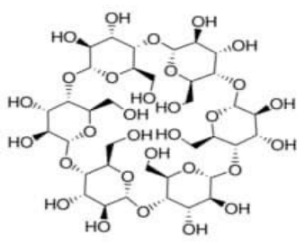	TCI *	10016-20-3	≥0.998
β -cyclodextrin	C_42_H_70_O_35_	* 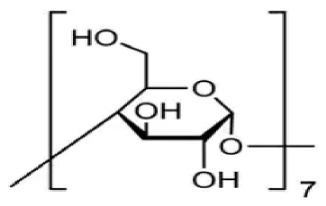 *	TCI *	7585-39-9	≥0.998
2-HP-β-cyclodextrin	C_66_H_112_O_42_	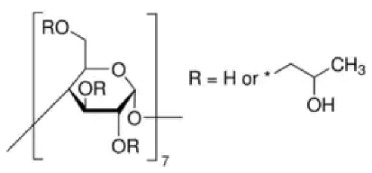	TCI *	128446-35-5	≥0.998
Sodium hydroxide micropills	NaOH		Avantor	1310-73-2	≥0.998

* TLC (Tokyo Chemical Industry).

## Data Availability

No applicable.

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
