# Peer review of "Stability of the Inclusion Complexes of Dodecanoic Acid with α-Cyclodextrin, β-Cyclodextrin and 2-HP-β-Cyclodextrin"

_molecules, 2023, doi:10.3390/molecules28073113_

Round 1
Reviewer 1 Report
This manuscript deals with the stability of the inclusion complexes of dodecanoic acid with α-cyclodextrin, β-cyclodextrin, and 2-HP-β-cyclodextrin. The stability was analyzed from the size of the cavity in cyclodextrins, their molar mass, and the structure of the fatty acid. The manuscript is overall well-prepared and interesting, and the data support the suggestion in the conclusions. I would recommend publication in the molecules after the author addresses the following points:
1. What is the theoretical background to present equation (15)? Please elaborate to describe the background with references.
2. Instead of the data listed in Table 3 and shown in Figure 4-6, it is better to present the functions with the values of the coefficients (A,B,and C).
3. The data in Table 2 include the range of each value. As long as the data listed in Table 3 were acquired from those in Table 2, each variable in Table 3 should be with the range.
4. At line 380 to 381, “But also because of the different amount of water molecules present inside the cyclodextrin molecules.” is ungrammatical expression.
5. The analysis of the size effect is little convincing for the minimum point of the free energy change. Why is the temperature of the minimum free energy higher at the α-cyclodextrin than those of the β-cyclodextrin and the 2-HP-β-cyclodextrin? In terms of the space, the β-cyclodextrin was closer to the α-cyclodextrin than the 2-HP-β-cyclodextrin. But in terms of the amount of the free energy change and the temperature of the minimum free energy, the β-cyclodextrin was closer to the 2-HP-β-cyclodextrin than the α-cyclodextrin. Does this analysis mean that the space is less significant for the free energy change? If then, what is the most significant for the free energy change?
6. The first abbreviation-CD- at line 29 should be expressed with full name.
Author Response
Answer for the Reviewer 1
I would like to thank you for the constructive comments and corrections of the manuscript to enhance its scientific values. In the revised version of my paper I included all comments and suggestions. I hope that the responses to the comments/suggestions presented below are sufficient and satisfactory. In the revised version of the work, all the changes are marked in yellow. Our responses to your comments are below.
- Reply to comment 1
Literature references [27-29] and the following sentence have been added to the main text of the article (according to the Reviewer's suggestion):
“The values of parameters A, B and C are calculated from the figures showing the changes of the Gibbs free enthalpy as a function of temperature and then they are inserted into the analyzed equation [27-29]. The values of these parameters for the studied in this paper systems are summarized in Table S5.”
- Reply to comment 2
According to the Reviewer's suggestion, the values of parameters A, B and C of Equation 8 have been added to the Supplementary Material in the form of Table S5. The courses of changes of thermodynamic functions and their courses of changes presented in Table 3 and Figures 4-6 are necessary for a full illustration of the interpretation of results carried out in the work.
- Reply to comment 3
In Table 2, the available literature Kf values for various carboxylic acids with a chain length of C8 - C10 have been intentionally summarized in order to show how diametrically different values of this constant are obtained depending on the research method used. Unfortunately, it is not possible to compare the values of Kf presented in my work with literature data because they are missing.
- Reply to comment
According to the Reviewer's suggestion included in point 4 of the review, the following sentence:
“The observed differences may result from a slightly different structure of the cyclodextrins discussed. But also because of the different amount of water molecules present inside the cyclodextrin molecules.”
has been changed to:
“The observed differences may result not only the different structure of the discussed cyclodextrins, but also from different amounts of water molecules which are located in the "covities" of the studied cyclodextrins.”
- Reply to comment 5
The free space in the cyclodextrin cavity certainly has a significant impact on the obtained ∆G0 values. However, it does not mean that it is the only factor affecting the value of this function. It should be noted that both temperature changes and the spatial structure of the tested cyclodextrins (affecting access to the cyclodextrin cavity) also play a very important role. It is difficult to answer on the question which of these factors is dominant in a given solution, and therefore we are dealing with the effects that the Reviewer pointed out.
- Reply to comment 6
At the Reviewer's suggestion, the names of the tested cyclodextrins given in the Introduction (line 29) in the form of the abbreviation a-CD, b-CD and 2-HP-b-CD have been replaced with their full names; a-cyclodextrin, b-cyclodextrin and 2HP-b--cyclodextrin.
The revised version of the article includes corrections to the English language recommended by a native English speaker.

Reviewer 2 Report
The study is an addition and extension of some previous works of the same author/group (e.g. Kinart, Z.; Tomaš, R. Molecules 2022, 27(14), 4420 & Kinart, Z. Molecules 2023, 28(1), 292.) There is nothing wrong with using an established technique, the conductometric method, and extending its applications to new topics, but there is no need to repeat the same content over and over again (e.g. mathematical equations, lines 138 - 206).
The introduction reads wordy with too many details. As a reviewer, I am not sure that this level of detail (e.g. lines 53-124) is justified. I strongly suggest that the author consider a revision to reduce the overall length of the introduction and improve readability. The current version lacks sufficient justification (text), emphasizing on why this proposed study is needed and can address the existing challenges.
Since the author has used the same technique, it would be worthwhile to compare different applications and highlight the changes and improvements in the discussion.
Author Response
Answer for the Reviewer 2
I would like to thank you for the constructive comments and corrections of the manuscript to enhance its scientific values. In the revised version of my paper I included all comments and suggestions. I hope that the responses to the comments/suggestions presented below are sufficient and satisfactory. In the revised version of the work, all the changes are marked in yellow. Our responses to your comments are below.
- Reply to comment 1
As suggested by the Reviewer, the equations 3 - 8 have been removed from the text. One explanatory sentence was added:
"The values of molar conductivity and values of the formation constant (Kf) of inclusion complexes were calculated according to the procedures described in our previous works [17, 18]."
- Reply to comment 2
According to the Reviewer's suggestion included in point 2 of the review, the following text was added to the chapter - "Conclusion":
“Taking into account and comparing the results presented in this work (as well as the results contained in our earlier works [17, 18]) with the literature data presenting the values of the constants for the formation of inclusion complexes of various carboxylic acids with cyclodextrins obtained by various measurement techniques (mainly spectral), it should be stated that that conductometry is the most accurate method used in this type of research.This research technique also allows for the study of the complexation process in a wide range of temperatures.”
- Reply to comment 3
I would like to thank the Reviewer for the comments and suggestions contained in point 3 of the review. In the revised version of this work the introduction has been shortened.
The revised version of the article includes corrections to the English language recommended by a native English speaker.

Round 2
Reviewer 1 Report
All of the issues have been addressed.